# Real-World Evaluation of Once-Weekly Subcutaneous Semaglutide in Patients with Type 2 Diabetes Mellitus in Spain (SEMA-RW Study)

**DOI:** 10.3390/nu16152545

**Published:** 2024-08-03

**Authors:** Irene Caballero Mateos, María Dolores García de Lucas, Viyey Kishore Doulatram-Gamgaram, Paloma Moreno-Moreno, Ana Isabel Jimenez-Millan, Manuel Botana-López, Juan Francisco Merino-Torres, Alfonso Soto-Gónzalez, José Carlos Fernández-García, Cristóbal Morales-Portillo

**Affiliations:** 1Endocrinology and Nutrition Department, Virgen Macarena University Hospital and Vithas Hospital, 41009 Sevilla, Spain; irenecaballerom@hotmail.com (I.C.M.); cr.morales@hotmail.com (C.M.-P.); 2Internal Medicine Department, Hospital Costa del Sol, 29603 Marbella, Spain; gdelucaslola@gmail.com; 3Department of Endocrinology and Nutrition, Regional University Hospital of Malaga, Biomedical Research Institute of Malaga (IBIMA), Faculty of Medicine, University of Malaga, 29016 Malaga, Spain; viyu90@hotmail.com; 4Endocrinology and Nutrition Department, Reina Sofía University Hospital, 14004 Córdoba, Spain; palomamoreno83@hotmail.com; 5Endocrinology and Nutrition Department, University Hospital, 11510 Puerto Real, Spain; ajmendo@gmail.com; 6Endocrinology and Nutrition Department, Lucus Augusti University Hospital, 27003 Lugo, Spain; manuelbotanal@gmail.com; 7Endocrinology and Nutrition Department, La Fe University Hospital, 46026 Valencia, Spain; merino_jfr@gva.es; 8Endocrinology and Nutrition Department, A Coruña University Hospital Complex, 15006 A Coruña, Spain; asotog10@yahoo.es

**Keywords:** type 2 diabetes mellitus, glucagon-like peptide-1 receptor agonist, once weekly semaglutide, glycemic control, body weight

## Abstract

Although, in randomized clinical trials, once-weekly subcutaneous semaglutide (OW s.c.) has demonstrated superior efficacy in comparison with placebo and active controls in terms of glycemic control and body weight reduction in patients with type 2 diabetes mellitus (T2DM), these results need to be confirmed in a real-world (RW) setting. An RW ambispective study (6 months retrospective and 6 months prospective) was conducted in 10 tertiary hospitals in Spain. We evaluated changes in HbA1c and body weight in patients with T2DM treated with semaglutide OW s.c. Additionally, we analyzed different subgroups of patients treated with semaglutide OW s.c. as an add-on to glucose-lowering therapy. A total of 752 patients with a mean age of 60.2 years, a mean HbA1c level of 8.5%, a mean body weight of 101.6 kg, and a mean T2DM duration of 10 years were included. At 12 months, compared with baseline, there was a mean difference of −2.1% in HbA1c levels (*p* < 0.001) and a mean difference of 9.2 kg in body weight (*p* < 0.001). Moreover, there were statistically significant differences (*p* < 0.001) between baseline and month 12 in both HbA1c and body weight in the four subgroups receiving semaglutide OW s.c. as an add-on to glucose-lowering therapy. Semaglutide OW s.c. was well tolerated, with gastrointestinal disorders being the most commonly reported side effects. In this RW study, 12 months of treatment with semaglutide OW s.c. in patients with T2DM was associated with significant and clinically relevant improvements in glycemic control and weight loss, regardless of the glucose-lowering therapy received, and the overall safety profile was positive.

## 1. Introduction

Type 2 diabetes mellitus (T2DM) is a chronic metabolic disorder frequently associated with severe macrovascular and microvascular complications [1]. Its prevalence, incidence, and burden on healthcare systems are steadily increasing worldwide [2], and by 2030, the number of people with diabetes in Europe is expected to rise to 40 million [3].

Since the management of T2DM is challenging and complex, treatment guidelines for T2DM recommend the selection of glucose-lowering pharmacological therapy, a tailored and multifactorial patient-centered approach, including individualized glycated hemoglobin (HbA1c) targets, reducing body weight, decreasing associated cardiovascular risk factors (CVRFs) and comorbidities, delaying or preventing complications, and improving patients’ quality of life [4,5,6].

Glucagon-like peptide-1 receptor agonists (GLP-1 RAs) are a well-established and effective class of T2DM drugs that mimic the effect of native GLP-1 by enhancing glucose-dependent insulin secretion, suppressing glucagon release, and reducing hepatic glucose production [7,8]. In addition, these drugs delay gastric emptying, reduce appetite, and have been shown to improve cardiovascular risk factors [9,10,11,12]. Compared to other glucose-lowering drugs, GLP-1 RAs are highly effective at improving glycemic control and reducing body weight and blood pressure while showing a cardioprotective effect and involving a low risk of hypoglycemia [8,13,14]. Consequently, the 2022 Consensus Report on the Management of Hyperglycemia in T2DM by the American Diabetes Association (ADA) and the European Association for the Study of Diabetes (EASD) recommends GLP-1 RAs as second-line therapy for the treatment of T2DM because of their organ protection benefits beyond improving glycemic control [4].

Semaglutide is a once-weekly (OW) subcutaneous (s.c.) human long-acting GLP-1 RA that was approved by the European Medicines Agency (EMA) in 2018 for the treatment of insufficiently controlled T2DM as an adjunct therapy to diet and exercise, either as a second-line monotherapy or as an add-on treatment [15]. In addition, a new once-daily oral formulation of semaglutide has been recently approved by the EMA for the treatment of T2DM [16].

Semaglutide OW s.c. for the treatment of T2DM has been extensively studied in the phase III randomized clinical trial (RCT) SUSTAIN program. In all these clinical trials, semaglutide at doses of 0.5 mg and 1 mg OW s.c. demonstrated superior and clinically meaningful reductions in HbA1c, body weight, and systolic blood pressure (SBP) compared with placebo and several active comparators, such as basal insulin glargine, sodium–glucose cotransporter-2 inhibitors (SGLT-2i), and other GLP-1 RAs [17]. Furthermore, in the SUSTAIN-6 trial, which was conducted in patients with T2DM at high cardiovascular risk, semaglutide OW s.c., compared with placebo, significantly reduced the risk of major adverse cardiovascular events (MACE)-3 composite cardiovascular endpoints [18].

However, despite the safety and efficacy of semaglutide OW s.c. in RCTs, insights into its real-world use in clinical practice are needed. Furthermore, the effect of semaglutide OW s.c., as an add-on to other glucose-lowering therapies in patients with T2DM, has not yet been evaluated in a real-world clinical setting.

Hence, in this study, we aimed to evaluate changes in HbA1c and body weight in patients with T2DM treated with semaglutide OW s.c. in clinical practice in Spain and to analyze different subgroups of patients treated with semaglutide as an add-on to glucose-lowering therapy, semaglutide added, respectively, to noninsulin monotherapy, double/triple noninsulin therapy, basal insulin therapy, and basal-bolus insulin therapy.

## 2. Materials and Methods

### 2.1. Design

This was an RW multicenter observational ambispective study conducted in 10 tertiary hospitals in Spain. We included patients with T2DM who were aged 18 years or older, who had started treatment with semaglutide OW s.c. between June and July of 2019 (the date of first commercialization of semaglutide in Spain) and July 2020, and who had been on treatment with semaglutide OW s.c. for at least 6 months prior to their inclusion in this study. In addition, at least one measurement of HbA1c, blood pressure, or body weight had to be available within ±1 month of the first semaglutide prescription.

Data were collected retrospectively from the patient’s medical records from the semaglutide treatment initiation date (baseline or visit 0) to the inclusion date (visit 1) and prospectively from the inclusion date (visit 1) to at least 6 months of follow-up (visit 2).

The study protocol was reviewed and approved by the Ethics Committee of the Hospital Universitario Virgen Macarena (Seville, Spain), and written informed consent was obtained from all the patients.

### 2.2. Study Endpoints and Outcome Measures

At baseline, the following study variables were collected: age; sex; relevant medical history and disease status (T2DM, diabetic retinopathy, dyslipidemia, chronic kidney disease, metabolic dysfunction-associated steatotic liver disease [MASLD], cardiovascular disease, and congestive heart failure); physical examination variables (blood pressure (mmHg), heart rate (beats per minute), waist circumference (cm), body weight (kg), and body mass index); analytical variables (fasting plasma glucose (mg/dL), HbA1c (%), albuminuria (mg/g), estimated glomerular filtration rate (eGFR) (mL/min/1.73 m^2^), total cholesterol (mg/dL), HDL-cholesterol (mg/dL), LDL-cholesterol (mg/dL), triglycerides (TGs) (mg/dL), uric acid (mg/dL), aspartate transaminase (AST) (U/L), alanine transaminase (ALT) (U/L), platelet count, and creatinine (mg/dL)); semaglutide prescription date; semaglutide dosage (mg/week), prior and concomitant treatment with GLP-1 RAs (excluding semaglutide), metformin, dipeptidyl peptidase 4 (DPP-4) inhibitors, glitazones, sulphonylureas and glinides, SGLT2 inhibitors, basal insulin, rapid insulin, angiotensin-converting enzyme inhibitors (ACE) inhibitors and angiotensin receptor blockers (ARBs), thiazides and loop-diuretics, beta-blockers, calcium channel blockers, alpha-adrenergic blockers, statins, proprotein convertase subtilisin/kexin type 9 (PCSK-9) inhibitors, fibrates, ezetimibe, anti-coagulants, and antiplatelet agents.

In addition, changes in the following variables were subsequently assessed at 6 (v1) and 12 months (v2): HbA1c, body weight, fasting plasma glucose, waist circumference, SBP, diastolic blood pressure (DBP), eGFR, albuminuria, LDL-cholesterol, and TG levels.

For the primary efficacy objective, two independent endpoints were considered in this study: a) change from baseline to the end of the study (EOS) in HbA1c (%) and change from baseline to EOS in body weight (kg). Similarly, for this primary analysis, five populations were considered based on the treatment modality: (1) total sample of patients, (2) semaglutide added to noninsulin monotherapy, (3) semaglutide added to double/triple noninsulin therapy, (4) semaglutide added to basal insulin therapy, and (5) semaglutide added to basal-bolus insulin therapy.

Secondary efficacy endpoints included (1) a composite objective of a 1% reduction in Hb1Ac, a body weight reduction of at least 3%, and the absence of severe hypoglycemia (either clinically documented or reported by the patient); (2) the percentage of patients with HbA1c values <7.5% and <7.0%; and (3) changes in waist circumference, SBP, DBP, TGs, and LDL-cholesterol.

For safety evaluation, adverse events and serious adverse events (SAEs) were collected from medical records during the retrospective phase. Moreover, the following safety information was also collected: renal function over time (eGFR and albuminuria), renal and cardiovascular safety, major adverse cardiovascular events (MACEs) and major adverse renal events (MARE), total insulin dose reduction in insulin-treated patients, and an assessment of the incidence and severity of hypoglycemia.

### 2.3. Statistical Analysis

Two principal analyses were performed corresponding to each of the variables (HbA1c and body weight), and each outcome was analyzed in four independent subgroups: semaglutide added to noninsulin monotherapy, semaglutide added to double/triple noninsulin therapy, semaglutide added to basal insulin therapy, and semaglutide added to basal-bolus insulin therapy. Therefore, an equal partition of the global alpha value (0.05) was performed for the eight independent confirmatory analyses, implying that the statistical tests were adjusted to a nominal significance level of <0.0064. Four hundred and six patients were necessary in this study in order to conduct a pre-post analysis using a paired Student’s *t* test and to detect a change with at least a small difference in the effect size (i.e., Cohen’s d of at least 0.2) with a statistical power of 90% and a significance level of 0.64%. Therefore, assuming that 40 patients would be lost to follow-up, the final number of enrolled patients was 677.

The primary analysis population consisted of all patients included in the study. Baseline demographic and clinical characteristics were descriptively summarized as the mean, standard deviation, or median and interquartile ranges for continuous variables and absolute frequency and proportion for categorical variables. Changes in continuous variables were estimated using paired *t* tests. *p* values less than 0.0064 were considered to indicate statistical significance. Similarly, McNemar’s test with the corresponding 95% confidence interval (95% CI) was used for paired binary data analyses. In addition, multiple linear regression analyses were performed to assess the predictive value of some independent variables (namely, age, gender, BMI, baseline HbA1c or baseline weight [depending on the model], and diabetes duration) on changes in HbA1c and body weight.

The mean changes from baseline for continuous variables were estimated through a visitwise analysis (i.e., missing data were not imputed), whereas for the binary variables, both visitwise analyses and analyses including all subjects (i.e., we performed a non-responder imputation) were performed. All the statistical analyses were conducted using IBM SPSS Statistics Release 20.0.0.

## 3. Results

Seven hundred and eighty-two patients with T2DM who received at least one semaglutide prescription between June 2019 and July 2020 were initially screened for the study. Seven hundred and fifty-two patients were ultimately included in the analysis. The reasons for exclusion are summarized in Figure 1. Six hundred and eighty-six and four hundred and seventy-two patients completed visits 1 (6 months) and 2 (12 months), respectively.

The mean age (±SD) was 60.7 ± 11.2 years, the median (IQR) T2DM duration was 11.00 (5.00, 17.00) years, and 355 (47.2%) patients were females. In addition, the HbA1c level was 8.5% ± 1.8, the body weight was 101.6 kg ± 19.8, and the SBP was 136.0 mmHg ± 18.0. Additionally, 67.8% and 81.8% of the participants had baseline serum HbA1c levels ≥ 7.5% and 7.0%, respectively.

The most frequently recorded comorbidities were dyslipidemia (n = 554, 73.7%), hypertension (n = 542, 72.1%), obstructive sleep apnea–hypopnea syndrome (n = 144, 19.1%), MASLD (n = 126, 16.8%), and ischemic heart disease (n = 121, 16.1%).

The following proportions of patients treated with different OW semaglutide doses were recorded at study inclusion: 511 (68%) with 0.25 mg, 154 (20.5%) with 0.5 mg, and 87 (11.6%) with 1 mg. In addition, 305 (40.6%) subjects were concomitantly treated with basal insulin, 173 (23.0%) with bolus insulin, 342 (45.5%) with metformin, 342 (45.5%) with SGLT2 inhibitors, and 251 (33.5%) with a GLP1 RA other than semaglutide and who had switched to semaglutide prior to their enrollment in the study. The baseline demographic and clinical characteristics of the study population are shown in Table 1, Table 2 and Table 3.

Treatment with semaglutide OW resulted in statistically significant reductions in HbA1c and body weight, which were the two independent coprimary endpoints of the study. At 12 months (v2) compared with baseline (v0), there was a mean difference of −2.1% in HbA1c levels (*p* < 0.001) (Figure 2A) and a mean difference of 9.2 kg in body weight (*p* < 0.001) (Figure 2B). Significant mean differences at six months (v1) versus baseline were also observed for both HbA1c (−2.0%) and body weight (−7.0 kg) (*p* < 0.001). The following variables were significantly associated with changes in HbA1c at the end of the study: baseline HbA1c (beta −0.88, 95% CI −0.93 to −0.84) and diabetes duration (beta 0.03, 95% CI 0.02 to 0.04). Likewise, some variables showed significant associations with changes in body weight at study end: age (beta 0.11, 95% CI 0.04 to 0.17), baseline BMI (beta −0.31, 95% CI −0.43 to −0.20), and baseline HbA1c (beta −0.50, 95% CI −0.86 to −0.14).

Additionally, a subgroup analysis assessing the effectiveness of patients previously treated with GLP1RA yielded the following results: changes in HbA1c (mean −1.19, 95% CI −1.34 to −1.04) and changes in body weight (mean −6.44, 95% CI −7.17 to −5.71).

Notably, the analyses of the four subgroups (Table 4) revealed statistically significant differences in the mean HbA1c and body weight between baseline and month 12 (*p* < 0.001).

At 12 months, 354 (47.1%) and 416 (55.3%) study participants achieved an HbA1c < 7.0% and an HbA1c < 7.5%, respectively. Differences from baseline values were significant in both groups (*p* < 0.001). Considering only the valid cases at 12 months (N = 468; visit-wise analysis), the proportion of patients who achieved glycemic control was 75.6% for HbA1c <7.0% and 88.9% for HbA1c < 7.5%.

A total of 86 of the 428 participants (20.1%) met the composite objective of a 1% reduction in Hb1Ac, a body weight reduction of at least 3%, and the absence of severe hypoglycemia at the end of the study (12 months).

A significant mean reduction of −6.2 mmHg in SBP at 12 months was observed (*p* < 0.001); mean changes in DBP (−1.3 mmHg) at the same follow-up time point were not significant. In addition, the mean reductions in the following study variables reached statistical significance at EOS: a) waist circumference (−7.5 cm; *p* < 0.001); b) fasting plasma glucose (−46.6 mg/dL; *p* < 0.001); TG (−53.5 mg/dL; *p* < 0.001); and LDL-C (−14.1 mg/dL; *p* < 0.001). No significant changes in eGFR, UAE, AST, or ALT were observed at 12 months.

Changes in the semaglutide dosage administered to a total of 750 evaluable patients during the study were as follows: At baseline, 544 (72.5%), 117 (15.6%) and 89 (11.9%) patients were treated with 0.25 mg, 0.5 mg, and 1 mg semaglutide, respectively; at 6 months, 4 (0.5%), 318 (42.4%), and 421 (56.2%) patients were treated with 0.25 mg, 0.5 mg, and 1 mg semaglutide, respectively; and at 12 months, 6 (0.8%), 50 (6.6%), and 395 (52.7%) patients were treated with 0.25 mg, 0.5 mg, 0.5 mg, and 1 mg semaglutide, respectively, and the drug treatment was stopped in 11 (1.5%) patients. In addition, information regarding the drug dosage for 274 subjects (36.5%) was missing at 12 months.

After 6 months of treatment with OW semaglutide, 170 (22.6%) patients reported at least one adverse event (AE), with an incidence > 0.1% (Table 5). The most commonly reported AE was gastrointestinal complaint (n = 72, 9.6%). Seven participants experienced ten episodes of severe hypoglycemia (either clinically documented or reported by the patient): one experienced three episodes, and another reported two episodes. Among the subjects reporting MACEs, one patient had unstable angina, one patient had a stroke, one patient experienced a heart attack, and one patient had to be hospitalized due to heart failure.

## 4. Discussion

The present RW study conducted in endocrinology units in Spain showed that treatment with semaglutide OW s.c. in patients with T2DM resulted in clinically relevant and statistically significant reductions in HbA1c levels and body weights after 12 months of treatment. In addition, the proportion of patients achieving glycemic control notably increased from baseline to the end of the study. Similar results were consistently observed in the four subgroups in which semaglutide was added to either noninsulin or insulin therapy. The secondary composite endpoints combining targets of reduction in HbA1c, body weight, and SBP and the absence of severe hypoglycemia were also met. Furthermore, in addition to body weight reduction, other beneficial changes in factors contributing to CV risk were found; waist circumference, SBP, LDL-cholesterol, and TGs exhibited statistically significant reductions at the EOS that were regarded as clinically relevant.

The improvements in HbA1c and body weight found in our study were relatively consistent with the results reported in the SUSTAIN clinical trial program, albeit larger [17]. The mean HbA1c and body weight reductions were −2.1% and −9.2 kg, respectively, in the SEMA-RW study, while in the SUSTAIN trials, the changes in HbA1c and body weight ranged between—1.1% and—1.8% and from—3.5 kg to—6.5 kg, respectively. These differences might be partly explained by the different baseline characteristics of the study populations, such as diabetes duration (normally less than 10 years in the SUSTAIN RCTs compared with a mean duration of 10 years in our study), baseline HbA1c (usually less than 8.5% in the SUSTAIN RCTs compared with a mean of 8.5% in our study), and body weight values (overall, less than 95 kg in the SUSTAIN RCTs compared with a mean of 100 kg in our study) [17]. Furthermore, the visit-wise approach used in this study contributed to these results.

The findings from the present study are in line with those of other RW studies conducted in Spain [19,20,21,22,23,24,25,26] and other countries [27,28,29,30,31,32,33,34,35,36,37,38,39,40], where HbA1c and body weight reductions ranged from 0.7 to 1.5%, and from 4 to 12 kg, respectively. The wide variability in these real-world studies might be due to the different studies’ populations and methodologies. Many of these studies included both GLP-1 RA-naïve patients and patients with diabetes who switched from GLP-1 RA to semaglutide OW s.c., a population that was specifically excluded from the SUSTAIN trials. Evidence from RW studies supports switching from a GLP-1 RA to semaglutide since it is associated with further and significant reductions in HbA1c and body weight. Interestingly, 33% of the patients enrolled in the present study switched from a GLP-1 RA other than semaglutide to semaglutide OW s.c.

Semaglutide was generally well tolerated in the present study. The most frequent AEs were gastrointestinal (9.6%), such as nausea, vomiting, and diarrhea, and they were mostly transient (7.6%). Hypoglycemia was reported in only seven patients (0.9%). Of note, there were no reports of pancreatitis or pancreatic cancer. 

Our study has certain limitations, as well as some important strengths. The limitations of this study include its observational design; therefore, confounding factors cannot be excluded. We did not assess the effect of other antidiabetic medications, such as SGLT2 inhibitors, on weight loss. We also did not use specific scales to evaluate the impact of semaglutide on quality of life. Moreover, we did not include a control group in this study. In addition, some variables, such as blood pressure, could not be properly measured or collected in certain centers owing to COVID-19 pandemic restrictions. To the best of our knowledge, this is the first RW study to include a prespecified subgroup analysis evaluating the effect of semaglutide OW s.c. as an add-on to other glucose-lowering therapies in patients with T2DM. Additional strengths of the study are its multicenter, nationwide design, and large sample size.

## 5. Conclusions

In our RW study, 12 months of treatment with semaglutide OW s.c. in patients with T2DM was associated with significant and clinically relevant improvements in glycemic control and weight loss (regardless of the glucose-lowering therapy concurrently received by the patients), along with a significant improvement in the CV risk profile and an overall positive safety profile.

## Figures and Tables

**Figure 1 nutrients-16-02545-f001:**
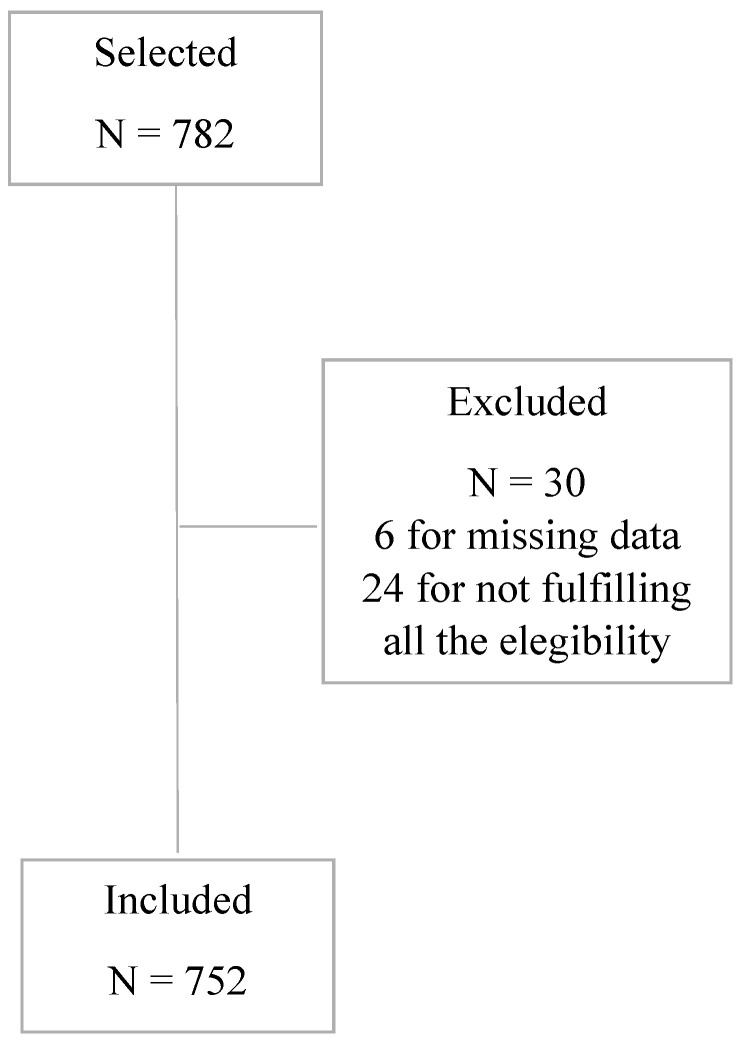
Flow diagram of patients’ disposition.

**Figure 2 nutrients-16-02545-f002:**
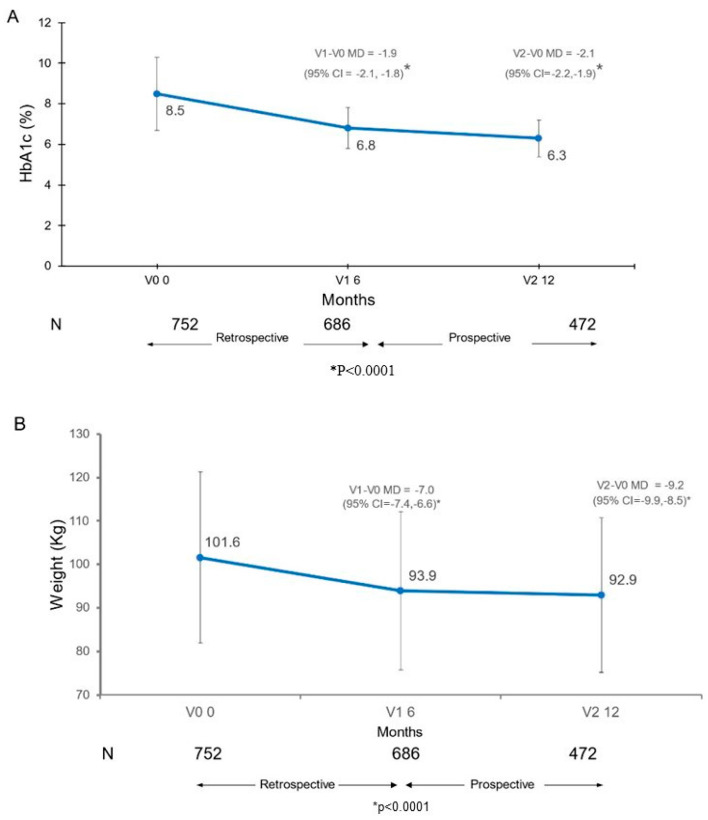
Change in (**A**) glycated hemoglobin (HbA1c), and (**B**) body weight (kg), over the study time. MD = mean difference; CI = confidence interval.

**Table 1 nutrients-16-02545-t001:** Baseline patient demographic and clinical characteristics.

Variable	
N	753
Age, years, mean (±SD)	60.71 (±11.18)
Female, n (%)	355 (47.2%)
Time since diabetes diagnosis, years, median (IQR)	11.00 (5.00, 17.00)
HbA1c, %, mean (±SD)	8.53 (±1.84)
HbA1c ≥ 7.5%, n (%)	510 (67.8%)
HbA1c ≥ 7.0%, n (%)	615 (81.8%)
Body weight, kg, mean (±SD)	101.62 (±19.77)
Height, cm, mean (±SD)	165.18 (±9.5)
Body mass index, kg/m^2^ mean (±SD)	37.11 (±6.55)
SBP, mm Hg, mean (±SD)	135.99 (±17.97)
DBP, mm Hg, mean (±SD)	78.18 (±11.18)
Waist circumference, cm, mean (±SD)	119.11 (±14.25)
Fasting plasma glucose, mg/dL, mean (±SD)	169.45 (±67.82)
Severe hypoglycemic episodes, n (%)	10 (1.3%)
Diabetic retinopathy, n (%)	98 (13.1%)
- Nonproliferative	67 (8.9%)
- Proliferative	17 (2.3%)
- Panretinal photocoagulated	14 (1.9%)
Hypertension, n (%)	542 (72.1%)
Dyslipidemia, n (%)	554 (73.7%)
CKD, n (%)	134 (17.8%)
OSAHS, n (%)	144 (19.1%)
NASH, n (%)	126 (16.8%)
IHD, n (%)	121 (16.1%)
Established cerebrovascular disease, n (%)	22 (2.9%)
PAD, n (%)	65 (8.6%)
CHF, n (%)	45 (6.0%)
Smoking status, n (%)	
- Current smokers	115 (15.3%)
- Former smokers	246 (32.7%)
Albumin, mg/dL, mean (±SD)	5.48 (±23.29)
Creatinine, mg/dL, mean (±SD)	1.27 (±5.59)
UAE, mg/dL, mean (±SD)	78.19 (±279.38)
eGFR, mL/min, mean (±SD)	81.21 (±24.52)
FIB4 score (0–6), mean (±SD)	1.24 (±0.65)
GOT, units/L, mean (±SD)	29.21 (±21.64)
GPT, units/L, mean (±SD)	34.10 (±29.13)
Blood platelet count, units/µL, mean (±SD)	1220.86 (±17,116.13)
Uric acid, mg/dL, mean (±SD)	13.63 (±190.10)
TC, mg/dL, mean (±SD)	173.94 (±41.4)
HDL-C, mg/dL, mean (±SD)	42.51 (±10.41)
LDL-C, mg/dL, mean (±SD)	95.17 (±35.85)
TG, mg/dL, mean (±SD)	213.41 (±234.121)

N: number; SD: standard deviation; IQR: interquartile range; HbA1c: glycated hemoglobin; SBP: systolic blood pressure; DBP: diastolic blood pressure; CKD: chronic kidney disease; OSAHS: obstructive sleep apnea-hypopnea syndrome; NASH: nonalcoholic steatohepatitis; IHD: ischemic heart disease; PAD: peripheral artery disease; CHF: chronic heart failure; UAE: urine albumin excretion; UACR: urine albumin-to-creatinine ratio; eGFR: estimated glomerular filtration rate; FIB4: Fibrosis-4 Index for Liver Fibrosis; GOT: glutamic-oxaloacetic transaminase; GPT: glutamic-pyruvic transaminase; TC: total cholesterol; HDL-C: high-density lipoprotein cholesterol; LDL-C: low-density lipoprotein cholesterol; TG: triglycerides.

**Table 2 nutrients-16-02545-t002:** Baseline patient laboratory test results.

Variable	N = 752
Creatinine, mg/dL, mean (±SD)	1.2 (5.5)
UAE, mg/dL, mean (±SD)	78.1 (±79.3)
eGFR, mL/min, mean (±SD)	81.2 (24.5)
TC, mg/dL, mean (±SD)	173.9 (41.4)
HDL-C, mg/dL, mean (±SD)	42.5 (10.4)
LDL-C, mg/dL, mean (±SD)	95.1 (35.8)
TG, mg/dL, mean (±SD)	213.4 (234.1)

N, number; SD, standard deviation; UAE, urine albumin excretion; eGFR, estimated glomerular filtration rate; TC, total cholesterol; HDL-C, high-density lipoprotein cholesterol; LDL-C, low-density lipoprotein cholesterol; TG, triglyceride.

**Table 3 nutrients-16-02545-t003:** Baseline medication.

Medication	
N	753
Semaglutide dose (s.c., OAW), n (%)	
- 0.25 mg	511 (68.00%)
- 0.5 mg	154 (20.50%)
- 1.0 mg	87 (11.60%)
Concomitant antidiabetic medication, n (%):	
- GLP1RA (excluding semaglutide)	251 (33.40%)
- Metformin	342 (45.50%)
- SGLT2 inhibitors	342 (45.50%)
- DPP4 inhibitors	157 (20.90%)
- Pioglitazone	15 (2.00%)
- Sulfonylureas	80 (10.60%)
- Insulin:	
○ Slow-acting insulin	305 (40.6%)
○ Rapid-acting insulin	173 (23.0%)
Other concomitant medication, n (%):	
- ACE inhibitors/ARAII	493 (65.60%)
- Beta blockers	173 (23.00%)
- Alpha blockers	29 (3.90%)
- CCB	196 (26.10%)
- Loop diuretics/thiazides	705 (93.80%)
- Potassium sparing diuretics	22 (2.90%)
- Statins	463 (61.60%)
- PCSK-9 inhibitors	2 (0.30%)
- Fibrates	52 (6.90%)
- Ezetimibe	86 (11.40%)
- Anticoagulants	53 (7.00%)
- Antiaggregants	234 (31.10%)

N: number; s.c.: subcutaneous; OAW: once a week; GLP1RA: glucagon-like peptide 1 receptor agonists; SGLT2: sodium–glucose cotransporter-2; DPP4: dipeptidyl-peptidase 4; ACE: angiotensin-converting enzyme; ARA II: angiotensin II receptor antagonists; CCB: calcium channel blockers; PCSK9: proprotein convertase subtilisin/kexin type 9.

**Table 4 nutrients-16-02545-t004:** Coprimary endpoints analyzed by subgroups of patients receiving semaglutide as an add-on treatment to different glucose-lowering therapies.

	Semaglutide Plus Noninsulin Monotherapy			Semaglutide Plus Noninsulin Double or Triple Therapy			Semaglutide Plus Basal Insulin Therapy			Semaglutide Plus Basal-Bolus Insulin Therapy		
	12-Months-Baseline Mean Difference (95% CI)	N	*p* Value *	12-Months-Baseline Mean Difference (95% CI)	N	*p* Value *	12-Months-Baseline Mean Difference (95% CI)	N	*p* Value *	12-Months-Baseline Mean Difference (95% CI)	N	*p* Value *
HbA1C (%)	−2.3 (−2.6, −2.0)	184	<0.001	−1.8 (−2.0, −1.6)	236	<0.001	−1.2 (−1.5, −0.9)	84	<0.001	−1.3 (−1.7, −0.8)	35	<0.001
Weight (kg)	−10.0 (−11.2, −8.7)	186	<0.001	−8.3 (−9.2, −7.4)	238	<0.001	−6.2 (−7.3, −5.2)	85	<0.001	−6.9 (−8.6, −5.3)	36	<0.001

HbA1c: glycated hemoglobin; CI: confidence interval. * Paired *t* test.

**Table 5 nutrients-16-02545-t005:** Side effects at V1 (6 months).

Adverse Event	N (%)
Severe hypoglycemia *^$^	7 (0.9)
Lower limb amputation	3 (0.4)
Gastro-intestinal overall	72 (9.6)
- Persistent	15 (2.0)
- Transient	57 (7.6)
MACE **	4 (0.6)
MARE ***	3 (0.4)
Retinopathy overall	4 (0.6)
- Newly onset	2 (0.3)
- Worsening	2 (0.3)

MACE: major adverse cardiovascular event; MARE: major adverse renal event. * Five patients had one episode, one patient had two episodes, and one patient had three episodes. ** Four specified MACEs: one case of unstable angina, one hospitalization for heart failure, one case of ictus, and one acute myocardial infarction. *** Specified MARE: 2 with new-onset microalbuminuria and 1 with dialysis. ^$^ Hypoglycemia: either clinically documented or reported by the patient.

## Data Availability

The original contributions presented in the study are included in the article, further inquiries can be directed to the corresponding author.

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
