# Peer review of "Real-World Evaluation of Once-Weekly Subcutaneous Semaglutide in Patients with Type 2 Diabetes Mellitus in Spain (SEMA-RW Study)"

_nutrients, 2024, doi:10.3390/nu16152545_

Round 1

Reviewer 1 Report

Comments and Suggestions for Authors

Thank you for the manuscript.

Please consider the comments below.

This is a real world evaluation study. The analysis was not adjusted for confounders - e.g. Age, BMI, HbA1c  at baseline. Would be useful for analysis to take this into account for example via a multivariate regression analysis.

What were the predictor of effective HbA1c and weight reduction.

Please provide information on how missing data was handled.

Was diabetes duration normally distributed? Usually its not and if so, to use median values.

What was the mean duration of follow up.

What was the relative effectiveness of patients whose GLP-1 was switched from a different glp-1 to Semaglutide?

Various real world studies have presented similar data (ref 27-40). How does this result compared with finds from other studies?

Author Response

Please see the attachment, thanks

Reviewer 2 Report

Comments and Suggestions for Authors

Dear Authors,

This is interesting manuscript, very well written, but I have a few suggestions/questions:

1. Introduction lines 91-95- this part is very difficult to read- maybe changing on: semaglutide added respectively to ..........,  to  ......

2 . Maybe in this place or in part statistical analysis (lines 154-156) name the groups for example group semaglutide with oral hypoglycemic agents (SEM+OHA), Samglutide aded to basal (SEM+BAS) etc. It will be better to read.

3. line 263- i suggest to change the gastrointestinal disorder to complaints. 

4. I suggest to add analysis of semaglutide effectiveness in this 4 groups with different treatment (in the manuscrit is only one sentence that effectiveness is better without insulin therapy)- with which treatment drug is the best effectiveness, maybe age, sex or weight of paricipants, maybe with lower additional disorders etc.

5. In side effects Authors mention amputations- I think it could be mentioned, but it is not side effect of semaglutide, I think.

6 Side effects- It should be mentioned that none of the patient have had the pancreatitis or cancer of pancrea- it is very important 

Author Response

Please see the attachment, thanks
